# Cross-Shelf Variation in Coral Community Response to Disturbance on the Great Barrier Reef

**Camille Mellin [1,2,*]**, **Angus Thompson [1]**, **Michelle J. Jonker [1]** and **Michael J. Emslie [1]**

[1]  Australian Institute of Marine Science, PMB No. 3, Townsville MC, Townsville, Queensland 4810, Australia; a.thompson@aims.gov.au (A.T.); m.jonker@aims.gov.au (M.J.J.); m.emslie@aims.gov.au (M.J.E.)

[2]  The Environment Institute and School of Biological Sciences, University of Adelaide, Adelaide, South Australia 5005, Australia

*  Correspondence: camille.mellin@adelaide.edu.au; Tel.: +61-(0)-8313-5432

**Abstract:** Changes in coral reef health and status are commonly reported using hard coral cover, however such changes may also lead to substantial shifts in coral community composition. Here we assess the extent to which coral communities departed from their pre-disturbance composition following disturbance (disassembly), and reassembled during recovery (reassembly) along an environmental gradient across the continental shelf on Australia's Great Barrier Reef. We show that for similar differences in coral cover, both disassembly and reassembly were greater on inshore reefs than mid- or outer-shelf reefs. This pattern was mostly explained by spatial variation in the pre-disturbance community composition, of which 28% was associated with chronic stressors related to water quality (e.g., light attenuation, concentrations of suspended sediments and chlorophyll). Tropical cyclones exacerbated the magnitude of community disassembly, but did not vary significantly among shelf positions. On the outer shelf, the main indicator taxa (tabulate *Acropora*) were mostly responsible for community dissimilarity, whereas contribution to dissimilarity was distributed across many taxa on the inner shelf. Our results highlight that community dynamics are not well captured by aggregated indices such as coral cover alone, and that the response of ecological communities to disturbance depends on their composition and exposure to chronic stressors.

**Keywords:** *Acropora*; benthic assemblages; chronic stressor; disassembly; disturbance; Great Barrier Reef; hard coral cover; reassembly

## 1. Introduction

Global warming is rapidly emerging as a universal threat to all ecosystems, reshuffling ecological communities into unprecedented assemblages [1]. Yet the effect of global warming does not occur in isolation, but is superimposed upon a background of chronic stressors (e.g., overfishing, reduced water quality) and acute disturbances (e.g., tropical cyclones, outbreaks of the corallivorous sea star *Acanthaster* spp.), most of which are being progressively exacerbated by climate change [2–4]. Ecosystem resilience can be defined as its capacity to absorb disturbances and maintain critical ecological functions and processes without fundamentally switching to an alternative stable state that is, for coral reefs, undergoing a phase shift from coral- to macroalgal-dominated communities [5,6]. However, the common practice of assessing resilience by grouping coral taxa together and measuring total coral cover obscures the specific response of different taxa to particular disturbances [6], preventing the identification of those that may win or lose under increasing disturbance regimes [1,7], and obfuscating how ecosystem processes and services might be expected to change in the future [8].

Despite the well-documented loss of half of the coral cover on Australia's Great Barrier Reef (GBR) over the past three decades [9], few studies have considered how coral community

composition changed post-disturbance (disassembly) and perhaps returned to its pre-disturbance state (reassembly) [1,10]. Recent findings on the GBR and elsewhere revealed that both the magnitude and the outcome of such processes are highly variable, with some communities successfully regaining their assemblage composition upon coral cover recovery [10,11], and others failing to do so [10,12]. The mechanisms underpinning such discrepancy remain unclear, mainly because research efforts have so far focused on a limited number of reefs and/or disturbance events, limiting the inferences that can be drawn from their results.

Spatial variation in the relative abundance of different taxa is also likely to influence community sensitivity to disturbance, because different taxa are associated with different life histories (e.g., competitive, stress-tolerant, weedy) that dictate how they respond to environmental variation [7,13]. Similarly, different types of disturbance affect coral taxa differently. For example, outbreaks of the crown-of-thorns sea star *Acanthaster* cf. *solaris* (CoTS) tend to target branching and tabular *Acropora* species while leaving massive species like *Porites* largely untouched. Likewise, there are winners and losers following coral bleaching [1,14] and cyclones. In addition to different mortality rates, among-taxa variation in coral recruitment following a disturbance is a major determinant of the winners and losers after multiple disturbances [15]. This means that the spatial distribution of different taxa is at least partly determined by natural variation in environmental conditions and that in the case of a disturbance, the response of different communities will also likely vary spatially, especially given that some disturbances may differ in severity across various habitats. On the GBR, location across the continental shelf represents a major source of variation in environmental conditions [16,17], in particular water quality [18,19], which influences coral growth rate [20] and selects for differing coral [21] and fish [16,21,22] assemblages. Therefore, such cross-shelf gradients are also likely to influence community disassembly and reassembly in response to disturbance.

Here, we compare the departure and recovery of coral communities from their 'pre-disturbance' baseline based on coral cover vs. community composition along the cross-shelf gradient. We defined pre-disturbance communities as those corresponding to the maximum coral cover observed at each reef (N = 46) over the 22-year study period (1996–2017). Specifically, our objectives were to: (i) characterize cross-shelf variation in community composition in the absence of disturbance; (ii) compare coral cover decline/recovery against community disassembly/reassembly across the GBR; (iii) identify the coral taxa that mostly contribute to the dissimilarity between disturbed and pre-disturbance communities; and (iv) disentangle the relative influence of acute and chronic stressors, and the composition of pre-disturbance communities, on community disassembly and reassembly following disturbance.

## 2. Materials and Methods

### 2.1. Survey Reefs

Australia's Great Barrier Reef (GBR) consists of more than 2900 individual reefs extending over 2300 km between 9 and 24° S latitude. Reef communities of the GBR have been monitored yearly between 1993 and 2005, and then biennially thereafter, by the Australian Institute of Marine Science's (AIMS) Long-Term Monitoring Program (LTMP) [23]. As part of the LTMP, benthic assemblages have been surveyed on 46 reefs in six latitudinal sectors (Cooktown-Lizard Island, Cairns, Townsville, Whitsunday, Swain and Capricorn-Bunker) spanning 150,000 km$^2$ of the GBR (Figure 1A). In each sector (with the exception of the Swain and Capricorn-Bunker sectors) at least two reefs were sampled in each of three shelf positions (i.e., inner-, mid- and outer-shelf).

### 2.2. Survey Methods

At each reef, three sites separated by >50 m were sampled within a single reef slope habitat (the first stretch of continuous reef on the northeast flank of the reef, excluding vertical drop-offs). Within each site, five permanently marked 50 m-long transects were deployed parallel to the reef crest, each separated by at least 10 m along the 6–9 m depth contour. Percent cover of benthic taxa were

estimated from digital images from each transect using point sampling of a randomly selected sequence of images [24]. The benthic organisms were identified to the lowest taxonomic resolution possible under five points arranged in a quincunx pattern in each image (n = 200 points per transect) and the data were converted to percent cover. The identified benthos components were then aggregated up to 54 benthic categories that included growth form and taxonomic resolution (species, genus and family) that were consistently applicable across the time-series. In this study, we focused on hard coral groups (N = 25; Table 1).

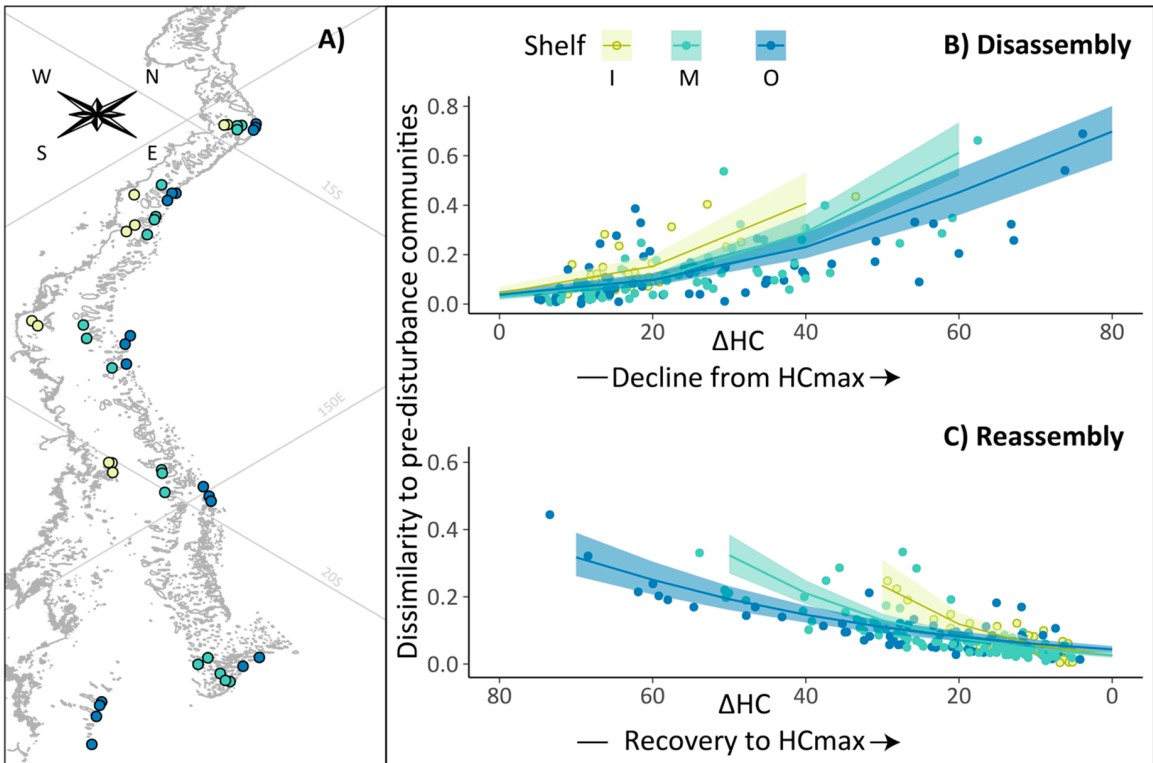

**Figure 1.** Coral community disassembly and reassembly across the continental shelf of Australia's Great Barrier Reef. (**A**) Location of survey reefs, (**B**) hard coral community disassembly following coral cover decline (N = 144) and (**C**) coral community reassembly following coral cover recovery (N = 207) on inner-shelf (I), mid-shelf (M) and outer-shelf reefs (O). On (**B**,**C**), the y-axis represents the Bray–Curtis distance (dissimilarity) between disturbed and pre-disturbance community composition. ΔHC (%) represents the difference between the maximum coral cover observed at each reef and the coral cover measured at that reef in any given year. Envelopes show 95% credible intervals from the posterior distributions estimated from the Bayesian hierarchical model.

**Table 1.** Hard coral groups surveyed and associated codes.

| Group Code | Group Description |
|---|---|
| ACBX | *Acropora* branching & bottlebrush |
| ACD | *Acropora* digitate |
| ACTO | *Acropora* tabulate & corymbose |
| COR_CBCF | non-*Acropora* coral branching & foliose |
| COR_CEMS | non-*Acropora* coral encrusting, massive & sub-massive |
| F_AGA | Agariciidae encrusting, foliose & submassive |
| F_DEN | Dendrophyllidae encrusting, foliose & massive |
| F_EUPH | Euphyllidae |
| F_FAV_CEMS | Faviidae encrusting, massive & sub-massive |
| F_FUN | Fungiidae |
| F_MER | Meruliniidae |
| F_MUS | Mussidae |

**Table 1.** *Cont.*

| Group Code | Group Description |
| --- | --- |
| F_OCU | Oculinidae |
| F_PEC | Pectiniidae |
| F_SID | Siderastreidae |
| G_AST | *Astreopora* |
| GE_ECH | *Echinopora* |
| G_GON_ALV | *Goniopora* & *Alveopora* |
| G_ISO | *Isopora* encrusting & sub-massive |
| G_MON | *Montipora* encrusting, foliose, massive & sub-massive |
| G_POC | *Pocillopora* |
| G_POR_B | *Porites* branching |
| G_POR_CEMS | *Porites rus*, *Porites* encrusting, massive & sub-massive |
| G_SER | *Seriatopora* |
| G_STY | *Stylophora* |

*2.3. Disturbance Data*

2.3.1. Chronic Stressors

We considered multiple indices of water quality at the survey reefs, including long-term (2003–2018) averages of near-surface chlorophyll-*a* concentration (*Chl a*; mg·m$^{-3}$), light attenuation at wavelength 490nm (*Kd490*; m$^{-1}$), and suspended sediment (non-algal particulates) concentration (*NAP*; mg·m$^{-3}$). These data were derived from MODIS satellite imagery and available through the eReefs Marine Water Quality Dashboard (http://www.bom.gov.au/marinewaterquality). Daily values were extracted from a 3 × 3 km square (9 pixels) adjacent to each reef and monthly means estimated. Our estimates represent the mean of these monthly means (to allow equal weighting across seasons with variable availability of daily estimates). We assessed the multicollinearity among water quality variables based on Spearman's ρ correlation coefficient, and excluded redundant ones (i.e., mostly correlated with other candidate predictors) from further analysis.

In addition, we included an index of reef accessibility based on distance to major human settlements that has previously been shown to negatively affect fish communities, with potential impacts on corals due to, e.g., anchor damage [25]. We also included the annual maximum Degree Heating Weeks [26] averaged between 1996–2016, reflecting spatial variation in chronic heat stress (in addition to marine heat waves leading to bleaching-induced mortality; see the Acute Stressors section below). Our rationale was that chronic heat stress is linked to sub-lethal deleterious effects on coral physiology, and there is evidence that coral assemblages exposed to such stress might have evolved through the selection of more heat-tolerant taxa or algal symbionts [27]. These variables were available at a 0.01° resolution [28] and we extracted, for each reef, values associated with the nearest grid node (distance ≤ 3 km).

2.3.2. Acute Stressors

The acute stressor data included two components (i) point-based records of coral damage collected concurrently with the benthic surveys and (ii) spatial layers of disturbance history and associated severity across the GBR assembled from various data sources.

(i) In point-based records of coral damage, disturbances were classified into five categories (i.e., coral bleaching, CoTS outbreaks, coral disease, cyclones or unknown) following Osborne et al. [29] based on visual assessment by experienced divers during broad-scale manta tow surveys around entire reef perimeters and intensive SCUBA surveys on fixed sites. A disturbance was recorded when the total coral cover decreased by more than 5% of its pre-disturbance value between two consecutive surveys. Each disturbance was identified by distinctive and identifiable effects on corals, such as the presence of CoTS individuals or feeding scars, or dislodged and broken coral indicative of

cyclone damage [29]. The additional category labelled 'unknown' was used to classify unidentified disturbances. This dataset thus resulted in a series of five binary variables coding the presence (1) or absence (0) of each type of disturbance in each year and at each reef where transect-based surveys of benthic assemblages were conducted.

(ii) Spatial layers of disturbance severity during the study period were compiled at a 0.01° resolution for coral bleaching, CoTS outbreaks and cyclones and are fully described in Matthews et al. [28]. Briefly, percent coral cover bleached was interpolated using inverse distance weighting (maximum distance = 1°; minimum observations = 3) from extensive aerial surveys at 641 reefs for the 1998, 2002 and 2016 mass bleaching events on the GBR [2,30]. Interpolated maps of CoTS densities were also generated by inverse distance weighting (maximum distance = 1°; minimum observations = 3) from the manta tow data collected by the LTMP in every year from 1996 to 2017 [31]. The potential for cyclone damage was estimated based on 4-km resolution reconstructed sea state as per Puotinen et al. [32]. This model predicts the incidence of seas rough enough to severely damage corals (top one-third of wave heights > 4 m) caused by cyclones for every cyclone between 1996–2016. We then used these spatial layers to associate the binary occurrence of tropical cyclones, CoTS outbreaks or coral bleaching (as per (i)) with its severity.

*2.4. Modelling*

2.4.1. Calculation of Disassembly and Reassembly

For each reef, we determined the year of maximum coral cover observed over the entire time series (1996–2017) and calculated the difference between this maximum and the coral cover observed in each subsequent year ($\Delta HC$), equivalent to the absolute change in coral cover used in recent studies [20]. We then calculated the Bray–Curtis dissimilarity between coral community composition observed in each year and in the year of maximum coral cover (thereby defining 'Disassembly' in the case of coral cover decline, 'Reassembly' in the case of coral recovery) (Figure S1). We first square-root transformed percent covers to reduce the influence of the most abundant groups.

2.4.2. Pre-Disturbance Communities

We defined, for each reef, pre-disturbance communities as those for which hard coral cover differed from its maximum value by less than 5% in absolute value (Figure S1). We quantified the composition of pre-disturbance communities based on a non-metric multidimensional scaling (nMDS) [33] with reefs as the observation units and average cover of the different hard coral groups in pre-disturbance years as the input data.

For each cross-shelf level, we identified the indicator taxa (i.e., hard coral groups) that characterized pre-disturbance communities based on the Dufrêne–Legendre index [34]. The Dufrêne–Legendre index identifies the taxa that are significantly more abundant in one group than in others; in our case, such groups were the three cross-shelf levels (inner-, mid-, outer-shelf). We subsequently quantified the contribution of each taxon to community disassembly or reassembly using a SIMPER analysis [35] based on the comparison of disturbed and pre-disturbance communities.

Finally, we assessed the variation in pre-disturbance community composition that was explained by the cross-shelf factor (*SHELF*), and that was explained by the different chronic stressors, using a permutational multivariate analysis of variance based on distance matrices (PERMANOVA) [36].

2.4.3. Hierarchical Linear Models

We first tested for any cross-shelf variation in disassembly (or reassembly) as coral cover changes by modelling disassembly (or reassembly) as a function of the $\Delta HC \times SHELF$ interaction using a Bayesian hierarchical linear model with a reef random effect (Equation (1)). To investigate possible causative factors without any confounding cross-shelf effects, we used a second model set without *SHELF*, but with fixed effects coding for the composition of pre-disturbance communities

(*COMM*; i.e., nMDS scores), chronic and acute stressors (Equation (2)). In this model set, the same set of chronic stressors were considered for each reef (i.e., irrespective of the year) while acute stressors referred to the disturbance(s) that preceded coral disassembly or reassembly in each year. In both model sets, we successively considered disassembly or reassembly as the response variable by splitting up the time series based on whether coral cover declined or recovered (Figure S1).

Bray–Curtis dissimilarity ($\delta$; reflecting either disassembly or reassembly) for the *j*th reef within the *i*th year was modeled as a proportion following a beta distribution of mean $\mu_{ij}$ and dispersion parameter $\phi$ (assumed constant over observations [37]; i.e., $\phi \sim U(0, 50)$ [38,39]) with a logit link function as follows:

$$\delta_{ij} \sim beta\left(\mu_{ij}\phi, \left(1 - \mu_{ij}\right)\phi\right)$$

with mean model (Equation (1)):

$$\ln\left(\frac{\mu_{ij}}{1 - \mu_{ij}}\right) = \alpha_j + \beta_{HC,j}.\Delta HC_{ij} + \beta_{SHELF,j}.\text{SHELF}_j + \beta_{INT,j}.\text{SHELF}_j \times \Delta HC_{ij} \quad (1)$$

or, in our second model set (Equation (2)):

$$\ln\left(\frac{\mu_{ij}}{1-\mu_{ij}}\right) = \alpha_j + \beta_{HC,j}.\Delta HC_{ij} + \beta_{COMM,j}.\text{COMM}_j + \beta_{CHRONIC,j}.\text{CHRONIC}_j + \beta_{ACUTE,j}.\text{ACUTE}_{ij} \quad (2)$$

and where the reef-level intercepts ($\alpha_j$) and slopes ($\beta_j$) followed a Gaussian distribution:

$$\alpha_j, \beta_j \sim N\left(0, \sigma_j\right)$$

$$\sigma_j \sim U(0, 100).$$

All covariates were standardized prior to modelling by subtracting their mean and dividing by one standard deviation. We assessed their relative effects based on the posterior distribution of their coefficients. Where the 95% credible interval of the posterior distribution overlapped zero, the effect was considered non-significant. We quantified each model's goodness of fit using an adaptation of R-squared to Bayesian models [40].

These models were fit using STAN via the rstan package [41] in R 3.5.1 [42], with inferences made from the Markov Chain Monte Carlo algorithm. Three parallel chains of 5000 iterations (including 500 for warm-up) were run to look for convergence of posterior parameter estimates using the Gelman–Rubin convergence statistic (R-hat) [43]; posterior traces and predictive intervals were also examined for evidence of convergence and model fit. All model diagnostics provided no evidence for lack of model fit.

## 3. Results

### 3.1. Community Disassembly and Reassembly

During disassembly (N = 144), the dissimilarity between pre- and post-disturbance communities increased as coral cover declined, and decreased again as coral cover increased towards its maximum value during reassembly (N = 207) (Figure 1). For both disassembly and reassembly, significant interactions between $\Delta HC$ and *SHELF* in our first model set (Figure S2) indicated these responses were amplified on the inner shelf. In other words, a similar difference in hard coral cover equated to greater dissimilarity in community composition on inner-shelf reefs than on mid- or outer-shelf reefs. Hierarchical linear models with the $\Delta HC \times SHELF$ interaction explained 78% and 86% in disassembly and reassembly respectively ($R^2$).

### 3.2. Effects of Pre-Disturbance Community Composition, Chronic and Acute Stressors

The nMDS mostly discriminated inner-, mid- and outer-shelf pre-disturbance coral communities along the first axis (MDS1) (Figure 2), which we used as a proxy for cross-shelf variation in community

composition in the second hierarchical model set. The PERMANOVA indicated that a total of 31.1% variation in the composition of pre-disturbance coral communities was explained by cross-shelf location (*P* < 0.001), and 28.1% by chronic stressors related to water quality (including 25.2% explained by *NAP* alone; *P* < 0.001). We found no effect of chronic heat stress or reef accessibility on the spatial variation in pre-disturbance community composition (PERMANOVA; *P* > 0.05).

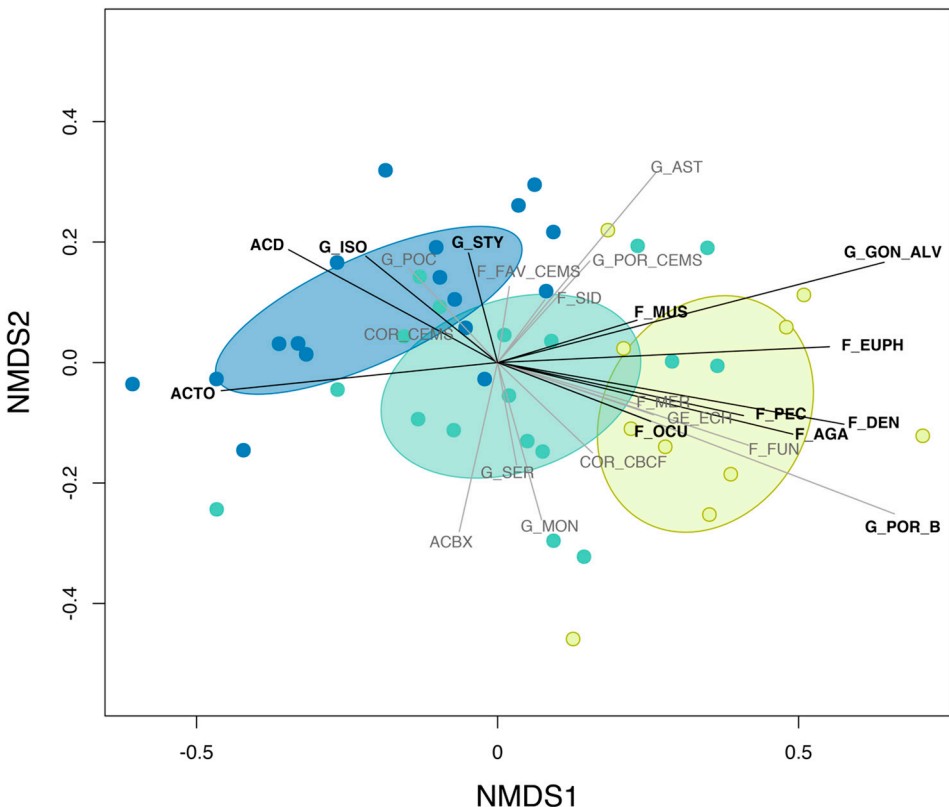

**Figure 2.** Non-metric multidimensional scaling of pre-disturbance communities at inner (yellow), mid (green) and outer (blue) shelf reefs (2D stress = 0.12). Dots show individual reefs, convex hulls delineate the different shelf assemblages (i.e., one standard deviation around the assemblage centroid) and segments show hard coral groups. Hard coral groups shown in black represent indicator taxa as per Figure 3. See Table 1 for hard coral group codes and description.

Given the multicollinearity among chronic stressors related to water quality (Figure S3), and between these stressors and pre-disturbance community composition, we only retained *Chl a* as a proxy for chronic stressors related to water quality in our second model set. *Chl a* was indeed the least correlated to coral community composition in pre-disturbance years (based on the PERMANOVA) and to *NAP* (main correlate of coral community composition based on the PERMANOVA), thus minimizing confounded effects of cross-shelf variation in both community composition and water quality variables.

Among all model predictors, Δ*HC* and the cross-shelf variation in community composition (with positive MDS1 values associated with inner-shelf communities) were the strongest predictors of dissimilarity between disturbed and pre-disturbance communities during both disassembly and reassembly (Figure 3). We found no significant effects of chronic stressors on community dissimilarity. Among acute stressors, only the severity of tropical cyclones had a significantly positive effect on community disassembly. Hierarchical linear models with community composition, acute and chronic stressors explained 79% and 78% of variation in disassembly and reassembly respectively ($R^2$). All posterior parameter estimates successfully converged (R-hat = 1; Figure S4).

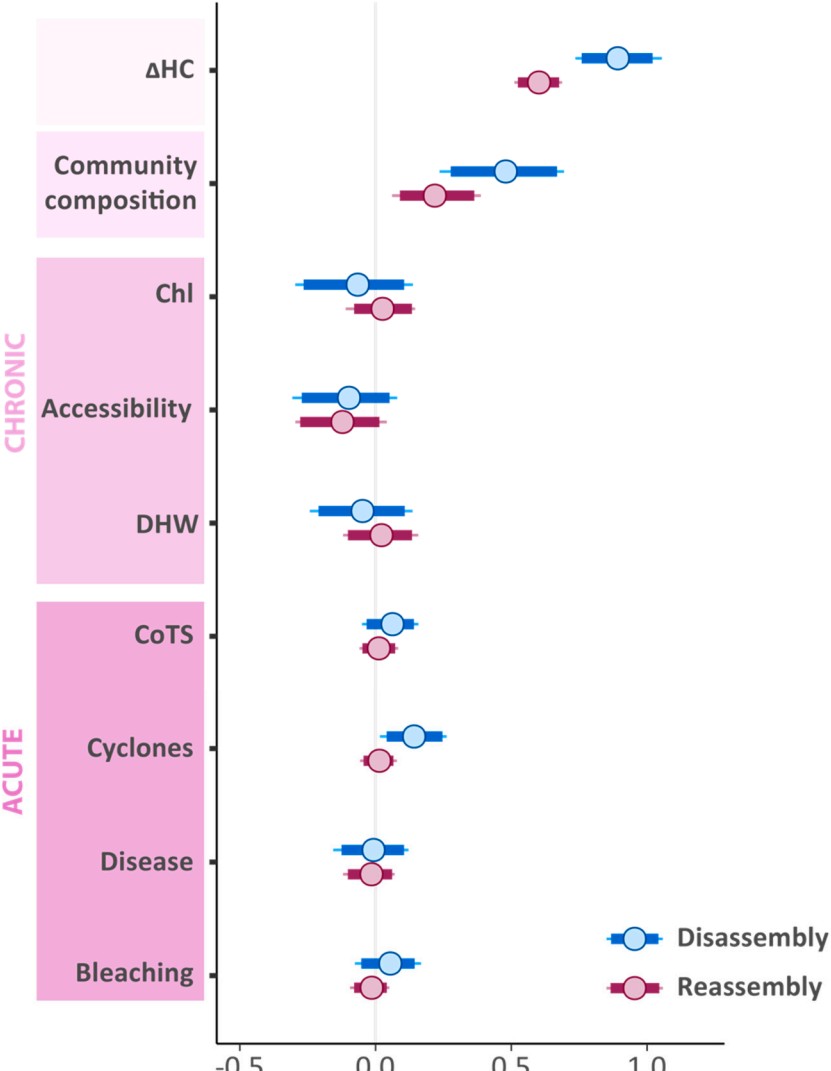

**Figure 3.** Posterior distributions of model coefficients (i.e., slopes) for fixed effects included in the Bayesian hierarchical models of community disassembly/reassembly following coral cover decline/recovery. Dots indicate the median, bars the 90% credible interval (CI) and whiskers the 95% CI. Where the 95% CI overlap zero, effects are considered non-significant. Community composition represents the first score of a non-metric multidimensional scaling of pre-disturbance communities (Supp. Figure 3). With Δ*HC*: difference from maximum coral cover; Chl-a: long-term average of chlorophyll-a concentration; DHW: Degree Heating Weeks; CoTS: outbreaks of the Crown-of-Thorns sea star (*Acanthaster* cf. *solaris*).

*3.3. Pre-Disturbance Community Composition and Taxon-Specific Contribution to Disassembly and Reassembly*

In pre-disturbance years, hard coral cover differed among shelf levels (Kruskal–Wallis test; *P* = 0.008) and was on average higher on inner-shelf reefs, despite greater maximum values recorded on the outer shelf, and mid-shelf to a lesser extent (Figure S5).

The composition of pre-disturbance coral communities differed across the shelf and was characterized by distinct indicator taxa (as identified by the Dufrêne–Legendre index) (Figure 4A). Inner-shelf communities were characterized by encrusting, foliose and submassive Agariciidae; and by Dendrophyllidae, Euphyllidae, Mussidae, Oculinidae, Pectiniidae, *Goniopora, Alveopora* and branching

*Porites*. Outer-shelf communities were characterized by *Acropora* (digitate, tabulate and corymbose), *Isopora* (encrusting and submassive) and *Stylophora*. No particular hard coral group characterized mid-shelf communities.

The contribution of each taxon to community disassembly or reassembly also differed across the shelf (Figure 4B). On outer shelf reefs, the main indicator taxa (tabulate and corymbose *Acropora*; ACTO) mostly contributed to community disassembly (or reassembly). This was not the case on inner shelf reefs, where contribution to disassembly (or reassembly) was more evenly distributed across taxa, and mostly attributed to non-indicator taxa of pre-disturbance composition such as branching and bottlebrush *Acropora* (ACBX), tabulate and corymbose *Acropora* (ACTO), and *Montipora* (G_MON).

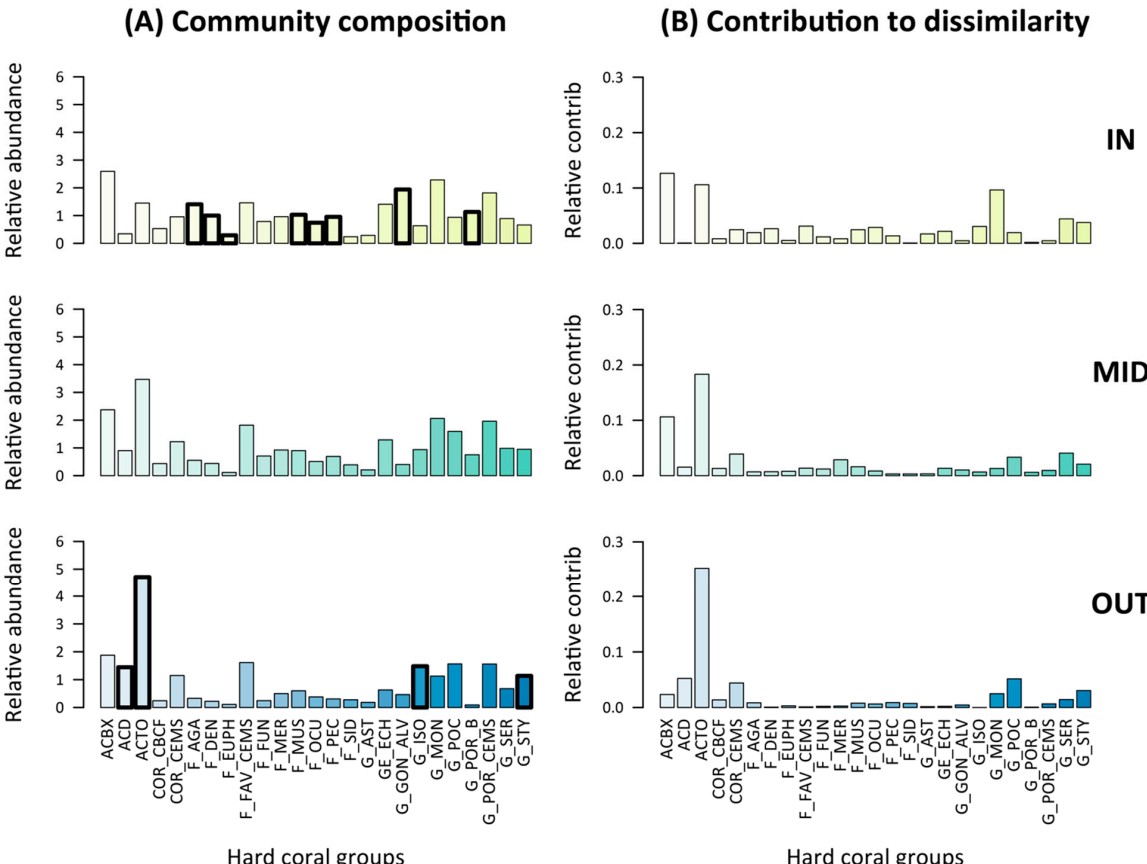

**Figure 4.** Composition of pre-disturbance hard coral communities (**A**) and relative contribution of each taxon to community dissimilarity (i.e., either disassembly or reassembly) following disturbance (**B**). Thick bar outlines in (**A**) show indicator taxa (Dufrene–Legendre index, *P* < 0.05) with, for inner-shelf reefs, F_AGA: encrusting, foliose and submassive Agariciidae; F_DEN: encrusting, foliose & massive Dendrophyllidae; F_EUPH: Euphyllidae; F_MUS: Mussidae; F_OCU: Oculinidae; F_PEC: Pectiniidae; G_GON_ALV: *Goniopora* and *Alveopora*; G_POR_B: branching *Porites*; and for outer-shelf reefs ACD: digitate *Acropora*; ACTO: tabulate and corymbose *Acropora*; G_ISO: encrusting and sub-masssive *Isopora*; G_STY: *Stylophora*. See Table 1 for other hard coral group codes and description.

### 3.4. Cross-Shelf Variation in Chronic and Acute Disturbance

Despite no significant effect on disassembly or reassembly based on our hierarchical linear model, the exposure to chronic stressors (except Degree Heating Weeks) was significantly higher on inner- than mid- or outer-shelf reefs (Kruskal–Wallis test, *P* < 0.001) (Figure 5A). We found clear cross-shelf differences for all water quality variables, reflecting the relatively high levels of suspended sediments and chlorophyll *a* in inshore waters, along with lower light availability. We also found clear cross

shelf differences for reef accessibility, a proxy for potential physical impacts from anchor damage and tourism activities.

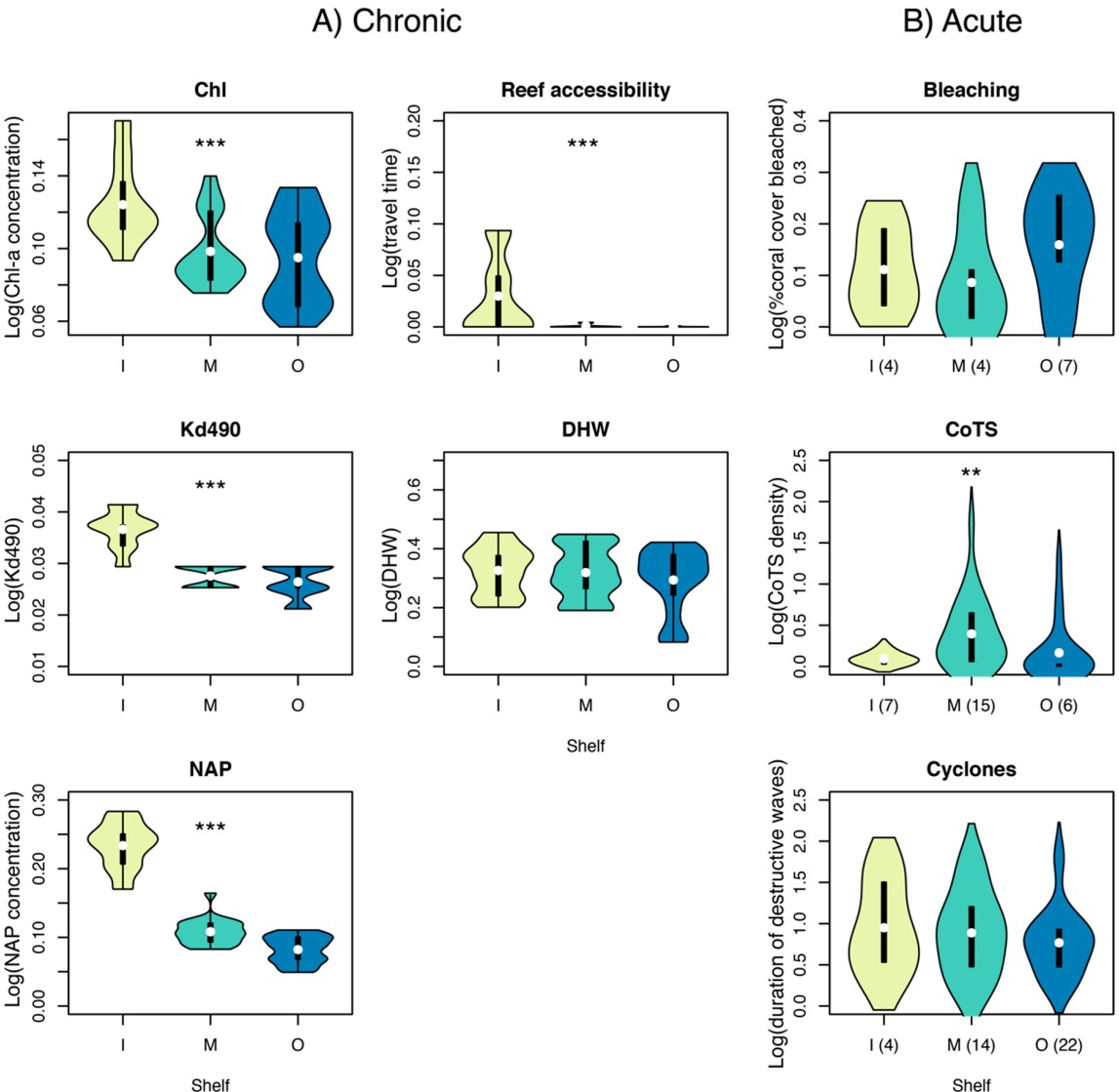

**Figure 5.** Distribution of chronic (**A**) and acute (**B**) stressors among inner (*I*), mid (*M*) and outer (*O*) shelf reefs. Violin plots represent data density, with plot width being proportional to the amount of data. Thick vertical lines represent the interquartile range, and the white dot the median. Asterisks show significant cross-shelf variation based on a Kruskal–Wallis test (***: $P < 0.001$; **: $P < 0.01$). For acute stressors, numbers in brackets represent the number of discrete disturbances. With Chl: long-term average of chlorophyll-a concentration; Kd490: light attenuation coefficient; NAP: non-algal particulates; DHW: Degree Heating Weeks; CoTS: outbreaks of the Crown-of-Thorns sea star (*Acanthaster* cf. *solaris*).

Of the acute stressors observed between 1996–2016 only CoTS densities differed statistically across the shelf (Kruskal–Wallis test, $P < 0.01$) with densities higher on mid shelf reefs (Figure 5B). Inner-shelf communities were exposed to relatively few, although severe, cyclones, while outer-shelf communities tended to be more severely bleached.

## 4. Discussion

Understanding why coral recovery has equated to community reassembly in some situations [10,11] but not others [10,12] requires a better understanding of the factors involved in community response to disturbance, and how they vary spatially. Our results show that, over the last 22 years, the same extent of coral cover decline (or recovery) did not equate to the same level of community disassembly (or reassembly) across the continental shelf of the GBR. As coral cover declined, community disassembly was exacerbated on inner-shelf reefs. Similarly, a greater level of coral recovery was necessary on the inner shelf for communities to reach the same level of reassembly as on the mid- or outer shelf. The similarity in both responses was to be expected given the symmetry of disassembly and reassembly processes [44]. These are important results, as the diversity of community responses to disturbance have remained elusive in most studies solely focusing on hard coral cover. By bridging this knowledge gap, our new findings will not only help ascertain the recovery potential of different hard coral communities and the underlying mechanisms, but will also inform about flow-on effects to other reef inhabitants such as invertebrates or fish that rely on hard corals for food or shelter [8,45].

The main predictor of both community disassembly and reassembly (other than the change in coral cover) was the composition of pre-disturbance communities. Most outer-shelf communities were characterized by tabulate *Acropora*, which was also the main taxon affected by disturbance. This suggests that, despite the occurrence of disturbance, the competitive dominance of tabulate *Acropora* led to community reassembly including a high proportion of this group. Conversely, on inshore reefs a wider range of taxa (including tabulate *Acropora*, among others) contributed to greater disassembly at the community level. This suggests a higher diversity of communities among inshore reefs (i.e., higher spatial turnover in community composition, as already demonstrated for coral reef fish [22]). Such higher spatial turnover in community composition on the inner shelf implies that dominant taxa (and those that mostly respond to disturbance) will likely vary among reefs and result in greater disassembly than on the outer shelf, where communities are more uniformly characterized by tabulate *Acropora* as the dominant taxa [46].

Up to 31% of variation in coral community composition was explained by reef position across the shelf, which is of the same order of magnitude as the 24% demonstrated previously from a subset of our dataset with slightly different and broader taxonomic groupings [21]. This cross-shelf gradient in community composition, with a greater dominance of Poritidae on inner-shelf reefs or *Acropora* spp. on mid- and outer-shelf reefs, reflected increasing water clarity and wave exposure from inshore to offshore reefs [21]. Similarly, in our study a total of 28% of spatial variation in community composition was explained by cross-shelf patterns in chronic stressors related to water quality, in particular the concentration of non-algal particulates, which also greatly differed among reefs. Although we cannot infer causation without uncertainty, this result corroborates the idea that greater and variable chronic stress on inshore reefs has led to spatially diverse communities composed of more stress-tolerant and slower-growing taxa (e.g., *Porites* spp.) [20,47]. Conversely, on outer-shelf reefs subject to less (and less variable) chronic stress, the most competitive and faster-growing taxa will likely dominate (e.g., tabulate *Acropora* spp. [48]) even in a regime of pulse disturbances at intervals of ten years or more [10]. Such life history strategies can predict species responses to disturbance and help identify species that may win or lose under increasing pressure from multiple stressors [7]. Species that are tolerant to multiple stressors are indeed expected to remain in the community as 'winners', whereas those that are sensitive to one or multiple stressors will likely be 'losers'. Our results suggest that, even though inner-shelf communities might be composed of taxa that are relatively more tolerant to chronic stressors related to water quality, some of these taxa might be more sensitive than others to the additional impact of acute stressors. For example, massive non-*Acropora* corals dominated inner-shelf communities with little contribution to disassembly following disturbances (suggesting they could be 'winners'), unlike branching *Acropora* that mostly contributed to disassembly despite lower relative abundance within the same communities (suggesting they could be 'losers'). The aftermath of the 2016 bleaching event revealed similar patterns, with a general shift from the dominance of branching

and tabulate species to depauperate assemblages dominated by taxa with simpler morphological characteristics and slower growth rates [1]. Such differential sensitivity among coral taxa under projected increase in cumulative pressure of chronic and acute stressors [20] might be expected to also increase the incidence of alternate stable states dominated by other benthic taxa such as macroalgae, sponges and soft corals [49], and reduced structural complexity that is likely to impact reef-associated organisms (such as fish or invertebrates) for which hard corals represent a key habitat [1,7,45].

Among the acute stressors included in our analysis, tropical cyclones caused greater coral community disassembly than expected based on the extent of coral cover loss. This result seems intuitive given the destructive nature of tropical cyclones that have the potential to dislodge even the least vulnerable corals (as opposed to CoTS outbreaks or bleaching that can result in partial mortality) [3]. Previous studies showed that cyclones were a major contributor to hard coral cover decline over the past decades [9,20,29]. Our results indicate that, in addition to the obvious impact on hard coral cover, tropical cyclones might exacerbate the community disassembly that is associated with hard coral loss per se. Conversely, that we found no effect of other acute stressors on disassembly or reassembly suggests that, based on our data, acute stressors other than cyclones did not exacerbate the dissimilarity in community composition that was expected under observed levels of coral decline. This might seem surprising in light of recent findings that demonstrated high levels of community disassembly in response to the severe 2016 bleaching event [1]. This discrepancy likely comes from the design of our study, focusing on cross-shelf gradients rather than latitudinal variation in both disturbance severity and community disassembly. Indeed, the impact of the 2016 bleaching event was the greatest on the northern section of the GBR [2], which might not have been picked up in our analysis grouping reefs of various latitudes into three cross-shelf positions. Another possible explanation is that the impact of the 2016 bleaching event has not been fully captured in the surveys yet. Future studies will be required to assess this impact and compare it to that of the 2017 event, which might be lower than expected based on the extent of heat stress, as most susceptible taxa died after 2016 [50].

Our results of greater disassembly on inshore reefs, derived from the 6–9 m depth zone, might extend to deeper reef habitats where lower light availability also favours the dominance of stress-tolerant species. Indeed, evidence suggests that community composition of deeper slope, mesophotic habitats might be more similar to inshore reefs than shallow offshore ones [51,52]. If this is the case, both deeper and more turbid environments might be associated with slower recovery, indicating that such patterns might be more common than previously assumed. More studies are required to assess the vulnerability of reef communities to disturbance in these less studied, yet widespread reef habitats.

Communities that regain coral cover but do not reassemble following disturbance are at risk of shifting to an alternative assemblage with altered ecosystem processes and function [10]. On inner-shelf reefs, a greater level of coral recovery was necessary for communities to reach the same level of reassembly than on mid- or outer-shelf reefs. Yet coral recovery also tends to be slower on the inner shelf, partly due to slower coral growth rate associated with a higher abundance of massive and slower-growing coral taxa [20]. This means that, in many cases, the time interval between disturbances would not suffice to allow for full coral recovery on inshore reefs, resulting in substantially lower levels of reassembly than on mid- or outer-shelf reefs. Johns et al. [10] showed that coral recovery took on average 9–11 years for three inner-shelf reefs, and 8–10 years for three mid-shelf reefs, with reassembly only occurring 3 years later or more. If another disturbance occurs in this interval, communities that have failed to regain their pre-disturbance composition will depart even farther from it. Given the predicted increase in the frequency of disturbances, a progressive drift to permanently modified coral assemblages is to be expected. Our results suggest that the pace of such drift will likely vary across the shelf based on environmental conditions and be exacerbated on inshore reefs dominated by stress tolerant, simpler growth forms with lower diversity. Some coral reefs have a considerable capacity to absorb recurrent disturbances and retain functional, albeit different, assemblages of coral

without undergoing a phase shift [6]. However, in a regime of increasingly frequent disturbances, the resilience of such communities (and the maintenance of their ecological function despite a drift of their taxonomic composition) will strongly depend on the factors that can promote resilience and alleviate the risk of shifting into an alternative, algal-dominated stable state. These factors include herbivory (and protection of herbivorous fish and invertebrates from fishing) [53,54], water quality [55,56] and, at a global scale, the mitigation of carbon emissions to curb the current rapid rate of global warming [2,57].

**Supplementary Materials:** The following are available online at http://www.mdpi.com/1424-2818/11/3/38/s1, Figure S1: Calculation of the Bray–Curtis distance to pre-disturbance communities from the 1996–2017 time series: example of the first three reefs in the dataset, Figure S2: Interaction plots for the hierarchical linear models, Figure S3: Multicollinearity among water quality indices, Figure S4: Trace plots for posterior parameter estimates, Figure S5: Distribution of hard coral cover in pre-disturbance years across shelf levels.

**Author Contributions:** Conceptualization, C.M. and M.J.E.; methodology, C.M.; formal analysis, C.M.; resources, A.T., M.J.J. and M.J.E.; data curation, A.T., M.J.J. and M.J.E.; writing—original draft preparation, C.M.; writing—review and editing, A.T., M.J.J. and M.J.E.

**Funding:** C.M. was funded by an ARC grant (DE140100701).

**Acknowledgments:** We thank members of the Australian Institute of Marine Science Long-Term Monitoring Program that have contributed to collection of the data used in these analyses, and B. Schaffelke for helpful comments on an earlier version of this manuscript.

**Conflicts of Interest:** The authors declare no conflict of interest. The funders had no role in the design of the study; in the collection, analyses, or interpretation of data; in the writing of the manuscript, or in the decision to publish the results.

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
