# Peer review of "Cross-Shelf Variation in Coral Community Response to Disturbance on the Great Barrier Reef"

_diversity, doi:10.3390/d11030038_

Round 1

Reviewer 1 Report

The comments are in pdf file.

Author Response

REVIEWER 1

The paper from Mellin et al. looks at the influences of diverse environmental drivers on the community disassembly and reassembly across shelf positions on the Great Barrier Reef (GBR). They found that coral communities situated on the inner-shelf tend to differ the most in term of responses to disturbances and recovery. Different statistical analyses allowed the authors to look at several aspects of community disassembly and reassembly patterns, which was nicely put together. However, I found that few important information and key results are missing which question the validity of some outputs and lessen the main statement of the study regarding the use of coral cover metric (L15-16, 22-25 and 320-325).

The authors use absolute changes in coral cover as the coral cover metric. There are pros and cons using absolute changes vs. others, mainly because changes are typically correlated with values of coral cover pre-disturbances. Changes are more important when they start from very high (or low) values of coral cover. It would be useful that the authors clearly state the rationale behind the choice of absolute changes in coral cover, especially in the context of shelf positions, where the percent of coral cover differs by nature.

RESPONSE: We agree with the reviewer about the pros and cons of using absolute change in coral cover, however the use of other metrics such as relative change also has pros and cons. For example, the use of relative change can be deceptive – a drop in coral cover from 10 to 5% is a 50% relative change, but does not mean the same as 80 to 40% from an ecological perspective. Therefore we chose to use absolute change, which is also in line with recent companion studies using the same data (e.g., MacNeil et al. in press) and have justified this choice in L161-162.

Please also note that we discounted very low absolute changes (<5%) as this has previously been shown to be within sampling error.

Another point, the number of absolute change measurements is higher during the reassembly period because coral cover recovery is slower than the decline (typically abrupt). I suspect that the number of Bray-Curtis dissimilarity values (calculated for all years and reefs) are different between the disassembly and reassembly periods. This information is missing in the manuscript whereas it contributes to set the background on how much data did the authors used to define disassembly and reassembly periods, for each shelf position. Indeed, a substantial discrepancy between the number of observations across shelf positions can influence the entire outcomes and interpretations of the study.

RESPONSE: The reviewer is right in that there are slightly more observations of reassembly (which is a slower process) than disassembly; however we disagree that it could change the direction or magnitude of the effect. A higher number of observations will only give more confidence (i.e. smaller credible intervals) in the relationship between absolute change in coral cover and dissimilarity in community composition.

We have now specified the number of observations for disassembly (N = 144) and reassembly (N = 207) in the results L216-218 and in the caption of Figure 1.

Several mistakes were found in the Bayesian hierarchical model section. The main one is about the prior distributions of model parameters. The dispersion parameter was not parameterized using a standard normal distribution because (1) it would be wrong to do it like this and (2) the authors contradict themselves as they state in the text that phi is constant over observations (L197). This means that phi was parameterized using a uniform distribution.

RESPONSE: We agree and apologize for this mistake; we have now corrected the assumption of phi following a uniform distribution, i.e. U(0,50) as suggested by Gelman (2006) and Figueroa-Zúñiga et al. (2013) (cited in the MS).

Also, the authors don’t show any hierarchy in the equations. In mathematics, the hierarchy is translated by the addition of extra-variance parameters for each model parameters.

RESPONSE: We have now added extra-variance parameters for each level of the hierarchy.

The authors showed a unique and fixed variance to 1 whereas the ranges of model parameters vary between 0 to 80 for delta HC, -0.5, 0.5 for Comm, 1 to 3 for SHELF and, 0-1 for the rest. I don’t see how the MCMC chains can converge using such restricted space.

RESPONSE: This is because all predictors were standardized prior to running the model to assist with interpretation of effect size; thus all model parameters are on the same scale with a fixed variance of 1. This is specified in L203.

It seems that the authors fit model parameters at the scale of reef (j), averaged across all the years. If so, the comma needs to be removed on the indexes. However, the overall distribution, at the scale of the GBR, is shown on Fig.2. Information explaining this step is missing in the method section. If this result comes from the model straight away, this mean that the authors have modelled the parameters such as:

Please revise the model equations.

RESPONSE: We calibrated the model for the jth reef within the ith year, with acute disturbances (and associated coral cover and community composition) varying among both reefs and years. We have therefore kept the comma on the indices and specified the hierarchy by adding an extra variance parameter (see above). The overall distribution, shown on figure 2, corresponds to the overall mean at the top of the hierarchy.

A visualization of model fitting the data is also missing in the SM. We have no information about how well the authors were able to fit the Bray-Curtis dissimilarity calculated from the data. This is very important to show, to make sense to the results of Fig. 2. Furthermore, the use of R2 to explain the % of variations in Bayesian models is not valid (Gelman et al. 2017). Please state how the R2 were estimated.

RESPONSE: We have now included posterior traces for all model parameters in the first model set and for each chain (N = 3) in the Supplementary Material. For the second model set, the traces show similar patterns; however showing them would be too cumbersome as this would result in 9 extra panels. We also specified that R-hat = 1 for all model parameters, confirming the convergence of posterior parameters estimates,

As for the R2, we used an adaptation developed by Gelman et al. (in press; now cited in the manuscript) for Bayesian models and implemented as the function ‘bayes_R2’ in the rstanarm package. We have now specified this in L206.

A potential lack for significant results for disturbance influences may come from the fact that the authors have ignored the temporal aspect of explanatory variables (which influence the pre-disturbance composition but also reassembly and disassembly processes as it has been widely shown in the literature). However, I understand that it is complicated to formulate when working with fragmented data. Maybe replacing the term “year” by “observations within period” will make the approach clearer.

RESPONSE: As explained above, we have now clarified that the models were calibrated for each reef within each year, and that the response variable (disassembly or reassembly) as well as most model parameters (i.e. community composition, acute disturbances) were temporally explicit (i.e. varying among years). This is specified in L192.

Minor comments:

I am not sure why the nMDS figure is in SM. This is one the main result that have been used in the hierarchical models and in conclusions. It also kinds of mirror the Figure 3 from the manuscript. However, it seems much more informative than Fig .3 and deserves to be in the manuscript.

RESPONSE: We agree and have now moved the nMDS to the main text (now Fig. 2).

L133-134: What about the category “Multiple disturbances”? It is shown in FigS1.

RESPONSE: This corresponds to the occurrence of more than once acute disturbance between two consecutive surveys and we have now clarified this in the caption.

Figure 1B: Should the sign the delta HC be negative? If not, the use of absolute difference needs to be state in the text.

RESPONSE: As explained above, this is now specified in L161.

Figure 1 caption and other places in the manuscript: The uncertainty band is named 95% credible intervals in the Bayesian framework. Please revise everywhere.

RESPONSE: Corrected throughout.

Reviewer 2 Report

The authors present an interesting and well analysed manuscript on disassembly and reassembly rules of coral communities on the GBR across environmental gradients. On a whole the paper is well written and clear, and will make a valuable contribution to Diversity. My only reservation about the study as currently written is the strawman argument of how “hard coral cover doesn’t capture community dynamics”. The abstract starts with:

“Hard coral cover is the most common metric of coral reef health and status, yet it does not inform on coral community composition”,

and the discussion begins with:

“These are important results, as the diversity of community responses to disturbance have remained elusive in most studies solely focusing on hard coral cover”.

While I don’t disagree with the importance of the results, the way the study is presented is a bit of a strawman argument: many (many) studies have looked at community level responses to disturbance on the GBR (e.g. Johns et al 2014). While hard coral cover approaches like the recent De’ath papers may be limited, I think there’s a better and more nuanced way of introducing the paper than to shoot down the strawman of “hard coral cover is limited”. That asides, the paper is well analysed and a valuable contribution to our understanding of GBR dynamics and the literature.

Minor comments:

Title: “Cross-shelf variation in coral community response to disturbance” – I’d add “on the Great Barrier Reef” or similar at the end. “cross-shelf” is mostly limited to the GBR in terms of terminology. Alternatively, state that it’s an environmental gradient or use “inshore to offshore” as more common terms. Considering the paper is framed around the concept of disassembly and reassembly of community structure, it would make it more compelling (and interesting) to use these terms in the title rather than (another) look at cross-shelf patterns.

Abstract: The abstract is confusing from the outset. Defining disassembly and reassembly would help here in an ecological context – does reassembly require recovery to initial starting assemblages?

This makes sense:

“Here we assess the extent to which coral communities disassemble as coral cover declines following disturbance, or reassemble during recovery, and how these patterns vary across the shelf on Australia’s Great Barrier Reef”

This doesn’t make sense in light of the preceding sentence. How does this change relate to disassembly or reassembly?

“We show that the same absolute change in coral cover equated to greater change in community composition on inshore reefs compared to mid- or outer-shelf reefs”

Use the definitions on line 45 and 46 in parenthesis to guide the abstract.

Main text:

Line 14: “across the shelf” isn’t immediately intuitive for a non-GBR researcher

Lines 337-338 or alternatively, tabular Acropora recovers more rapidly than other taxa

Lines 347-349 Worth citing Terry Done’s seminal 1981 paper here, as this finding is over three decades old.

Lines 374-375 why would reshuffling increase the incidence of altered states above and beyond simple loss of corals resulting in increased space availiblity?

Lines 403-404 again, the altered state component is unclear – increased turbidity may make it less likely to observe a switch to phototrophs due to reduced light levels

Lines 410-412 also reduced recruitment rates, see Sammarco’s early studies

Line 420: Interesting point. Can you speculate what this state may be?

Author Response

REVIEWER 2

The authors present an interesting and well analysed manuscript on disassembly and reassembly rules of coral communities on the GBR across environmental gradients. On a whole the paper is well written and clear, and will make a valuable contribution to Diversity. My only reservation about the study as currently written is the strawman argument of how “hard coral cover doesn’t capture community dynamics”. The abstract starts with:

“Hard coral cover is the most common metric of coral reef health and status, yet it does not inform on coral community composition”,

and the discussion begins with:

“These are important results, as the diversity of community responses to disturbance have remained elusive in most studies solely focusing on hard coral cover”.

While I don’t disagree with the importance of the results, the way the study is presented is a bit of a strawman argument: many (many) studies have looked at community level responses to disturbance on the GBR (e.g. Johns et al 2014). While hard coral cover approaches like the recent De’ath papers may be limited, I think there’s a better and more nuanced way of introducing the paper than to shoot down the strawman of “hard coral cover is limited”. That asides, the paper is well analysed and a valuable contribution to our understanding of GBR dynamics and the literature.

RESPONSE: We have now reworded the corresponding sections in the abstract and in the introduction to remove the strawman argument (L11-12 and 103-104).

Minor comments:

Title: “Cross-shelf variation in coral community response to disturbance” – I’d add “on the Great Barrier Reef” or similar at the end. “cross-shelf” is mostly limited to the GBR in terms of terminology. Alternatively, state that it’s an environmental gradient or use “inshore to offshore” as more common terms. Considering the paper is framed around the concept of disassembly and reassembly of community structure, it would make it more compelling (and interesting) to use these terms in the title rather than (another) look at cross-shelf patterns.

RESPONSE: We have now added ‘on the Great Barrier Reef’ to the title, however we feel that any rewording of ‘cross-shelf’ would be cumbersome – furthermore, ‘cross-shelf’ resonates better with the special issue this paper is part of (‘Cross-shelf Variation in the Structure and Function of Coral Reef Assemblages’). As for the concept of disassembly and reassembly, both terms are encompassed within ‘community response to disturbance’, which we prefer to use before defining the former in the main text.

Abstract: The abstract is confusing from the outset. Defining disassembly and reassembly would help here in an ecological context – does reassembly require recovery to initial starting assemblages?

RESPONSE: We have added the definitions for disassembly and reassembly in L13-14: “the extent to which coral communities departed from their pre-disturbance composition following disturbance (disassembly), and reassembled during recovery (reassembly)”.

This makes sense:

“Here we assess the extent to which coral communities disassemble as coral cover declines following disturbance, or reassemble during recovery, and how these patterns vary across the shelf on Australia’s Great Barrier Reef”

This doesn’t make sense in light of the preceding sentence. How does this change relate to disassembly or reassembly?

“We show that the same absolute change in coral cover equated to greater change in community composition on inshore reefs compared to mid- or outer-shelf reefs”

Use the definitions on line 45 and 46 in parenthesis to guide the abstract.

RESPONSE: As explained above, we have now clarified and reworded this in L13-14.

Main text: 

Line 14: “across the shelf” isn’t immediately intuitive for a non-GBR researcher

RESPONSE: We have now introduced the term in the abstract and feel it was adequately introduced within the introduction to make sense to a broad audience from there on.

Lines 337-338 or alternatively, tabular Acropora recovers more rapidly than other taxa

RESPONSE: We agree and have revised the relevant sentences to simplify the language and emphasis the Reviewer’s point.

Lines 347-349 Worth citing Terry Done’s seminal 1981 paper here, as this finding is over three decades old.

RESPONSE: We suspect the reviewer was referring to Done, T.J., 1982. Patterns in the distribution of coral communities across the central Great Barrier Reef. Coral Reefs, 1(2), pp.95-107, and we have now added this reference.

Lines 374-375 why would reshuffling increase the incidence of altered states above and beyond simple loss of corals resulting in increased space availiblity?

RESPONSE: We have revised this point for clarity – however note that it is expanded further in the concluding paragraph.

Lines 403-404 again, the altered state component is unclear – increased turbidity may make it less likely to observe a switch to phototrophs due to reduced light levels

RESPONSE: We have now deleted the ‘altered state’ component from this sentence to focus on recovery only.

Lines 410-412 also reduced recruitment rates, see Sammarco’s early studies

RESPONSE: Sammarco recorded recruitment between 3 and 15m with higher recruitment at 15m depths. This depth on mid and outer shelf is shallower than the more mesophotic reefs we are referring to as per the Bridge reference. We have added the term ‘mesophotic’ to make this explicit.

Line 420: Interesting point. Can you speculate what this state may be?

RESPONSE: We have reworded this sentence tothe pace of such drift will likely vary across the shelf based on environmental conditions and be exacerbated on inshore reefs dominated by stress tolerant, simpler growth forms with lower diversity’.

Reviewer 3 Report

Overall this is a very interesting and well executed study. The amount of data is impressive! I feel that it would strengthen the manuscript to include information about the relative diversity of the reefs studied before disassembly. The authors touched on this in the discussion, but I would like to see more discussion of it there and added to the introduction. White et al., 2017 found some interesting results comparing sites on an upper mesophotic reef that started with different levels of diversity.

My only other minor comment is that perhaps you should use "sea star" instead of "starfish"

Author Response

REVIEWER 3

Overall this is a very interesting and well executed study. The amount of data is impressive! I feel that it would strengthen the manuscript to include information about the relative diversity of the reefs studied before disassembly. The authors touched on this in the discussion, but I would like to see more discussion of it there and added to the introduction. White et al., 2017 found some interesting results comparing sites on an upper mesophotic reef that started with different levels of diversity.

RESPONSE: We agree with the reviewer that information on the relative diversity of each reef would be interesting, however given the taxonomic level at which corals were described, such diversity indices would be meaningless or at least difficult to interpret. On most reefs before disassembly, all of the hard coral groups were be represented but in different proportions – this is what we showed in Fig. 3a and corresponding analyses.

My only other minor comment is that perhaps you should use "sea star" instead of "starfish"

RESPONSE: Corrected throughout.